# REGION-AWARE INSTANCE CONSISTENCY LEARNING FOR APEX-FREE MICRO-EXPRESSION RECOGNITION

## ABSTRACT

Micro-expression Recognition (MER) is challenging due to the subtle motion. Existing methods heavily rely on the onset/apex pair to capture the most discriminative motion clues. This paradigm struggles with labor-intensive apex annotation and effective utilization of data. In this paper, we propose a novel apex-free paradigm for MER that eliminates the need for expensive apex annotations while effectively capturing subtle motion dynamics. Our key insight is that frames within the sequence exhibit spatially consistent and intensity varied motion cues relative to the onset frame. Motivated by this, our method treats each sequence as a set of multiple onset/near-median motion instances. To fully exploit weaker motion information conveyed by these diverse instances, our framework introduces an Instance Region Consistency (IRC) module that enforces visual attention consistency on similar facial activation regions across different instances within the same set. Furthermore, we present a Multi-Region Discovery (MRD) module with self-supervised learning to expand attention on more subtle activation regions which are typically neglected. Extensive experiments on four public micro-expression datasets demonstrate that our proposed approach surpasses state-of-the-art methods without using any apex frame annotations.

## 1 INTRODUCTION

Micro-expressions (MEs) are transient and involuntary facial movements reflecting genuine emotions that individuals attempt to conceal (Ben et al., 2021). Owing to this property, the ME recognition (MER) task has demonstrated crucial potential in various applications, e.g., lie detection (O'sullivan et al., 2009) and mental health assessment (Endres & Laidlaw, 2009). Despite recent progress, MER remains challenging since MEs cover facial activation regions with imperceptible intensity and brief duration.

Existing MER methods typically regard the motion between the onset and apex frames as the fundamental cue of representing subtle ME movements. The motion is commonly estimated using optical flow or deep learning-based methods, followed by a well-designed classification model that maps the input motion to emotion labels. Although the apex frame captures the most obvious motion cues for classification, accurately identifying the apex frame requires human coders to scan the ME sequence frame-by-frame manually, which is time-consuming and demands expert knowledge. Moreover, due to the difficulty in collecting and annotating ME data, the amount of data in current ME datasets is relatively small. Therefore, relying solely on onset/apex pairs not only limits practical usage of ME data, but also increases the risk of overfitting to specific facial activation regions. While recent self-supervised methods like AVF-MAE++ (Wu et al., 2025) have been proposed to learn general facial representation to mitigate overfitting, they typically require pre-training on external large-scale datasets, incurring substantial computational costs. Therefore, learning robust and general representations directly from small-scale ME datasets remains an open and challenging problem.

In this paper, we rethink the necessity of apex frames and propose an apex-free paradigm with Region-aware Instance Consistency Learning (Ra-ICL) for MER. Our empirical observations reveal that the effective motion information is not exclusively contained in the apex frame. As shown in Figure 1, we find that frames within the sequence exhibit motion patterns that share consistent spatial activation regions compared to the onset frame, while the motion intensity across these activation regions varies over time. This combination of spatial stability and intensity variation motivates us

to treat alternative pairs (e.g., onset vs. near-median) as carriers of weaker yet valid motion cues. Specifically, we propose to form instance sets composed of multiple motion instances from onset/near-median pairs for each sequence, rather than only one onset/apex pair.

To leverage diverse but weaker motion information delivered by these instances, we introduce an Instance Region Consistency (IRC) module to effectively capture subtle motion dynamics. This module is built on the observation that instances within the same set share the same emotion label and tend to activate similar facial activation regions. We randomly sample two instances to form a positive pair for each sequence and feed them through a Siamese network, obtaining their respective attention heatmaps via class activation mapping (Zhou et al., 2016). We then enforce visual at-

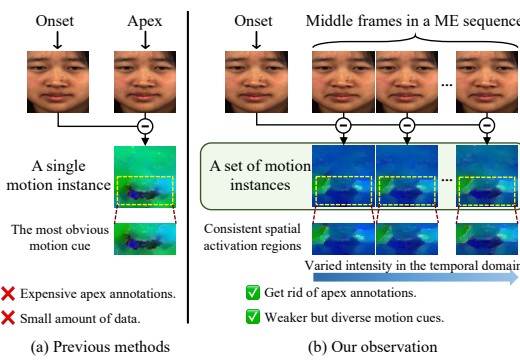

Figure 1: A ME sequence can be treated as a set of motion instances from multiple onset/near-median frame pairs without apex annotations. These instances exhibit consistent spatial activation regions with weaker but diverse motion cues.

tention consistency (Guo et al., 2019) between the two attention heatmaps to encourage the model to focus on similar facial activation regions across different instances. Due to the inherently low activation intensity of certain regions in MEs, attention heatmaps are prone to focus on prominent areas and neglect these subtle yet important regions. To address this issue, we further propose a Multi-Region Discovery (MRD) module inspired by recent advances in self-supervised facial representation learning (Gao & Patras, 2024). The MRD module uses a set of learnable facial queries to discover more subtle but meaningful facial regions in a self-supervised manner. Consequently, the attention of the model is expanded to encompass more subtle motion patterns.

To summarize, our main contributions are as follows:

- We rethink the necessity of apex frames and propose a novel framework Ra-ICL for apex-free MER. The ME sequence is represented by a set of multiple motion instances instead of a single onset/apex motion instance.

- We propose an IRC module to effectively capture subtle motion cues by enforcing visual attention consistency across different motion instances from the same set.

- To further enhance the perception of subtle activation regions, we present a MRD module to discover more meaningful facial regions with self-supervised learning, preventing decisions from localized activation regions.

- Extensive experiments on four public ME datasets demonstrate the effectiveness of the proposed method.

## 2 RELATED WORKS

**Motion Features for Micro-expressions.** In early works for MER, several texture-based feature descriptors were carefully designed to extract spatio-temporal features of ME motion, including LBP-TOP (Zhao & Pietikainen, 2007), HOG (Li et al., 2017), etc. Recently, researchers have adopted optical flow to characterize subtle facial muscle deformation of MEs. Optical flow serves as a robust representation of pixel-level inter-frame motion, effectively capturing both the magnitude and direction of motion. Some variants of optical flow were further proposed to offer more effective and robust motion features, such as MDMO (Liu et al., 2015) and Bi-WOOF (Liong et al., 2018). Since the apex frame reaches the maximum intensity of the ME motion, the optical flow between the onset and apex frames was typically utilized to represent observable ME motion features.

**Deep Networks for Micro-expression Recognition.** Existing methods typically leverage deep neural networks to extract emotion-relevant information from motion features between onset and apex frames. OFF-ApexNet (Gan et al., 2019) and Dual-Inception (Zhou et al., 2019) fed horizontal and vertical optical flow into a two-stream convolutional neural network (CNN) for feature enhancement, while STSTNet (Liong et al., 2019) further computed optical strain to form the triple

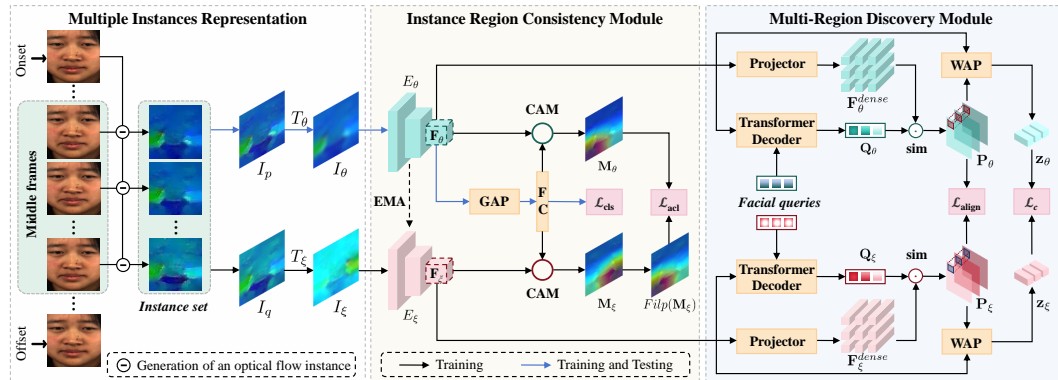

Figure 2: The framework of the proposed Ra-ICL. Given an input sequence, we generates multiple optical flow instances to form an instance set using onset/near-median frame pairs. The Instance Region Consistency (IRC) module takes random instance pairs from the instance set as input to learn effective motion information from weak yet valid motion cues, which enforces attention consistency on similar activation regions. A Multi-Region Discovery (MRD) module is further utilized to discover meaningful facial regions in a self-supervised manner, thus enhancing perception on subtle activation regions. Note that some intermediate results in MRD are omitted for brevity.

stream of a shallow 3D-CNN. In addition to CNN-based methods, Zhang et al. (2022a) first proposed a purely Transformer-based (Vaswani et al., 2017) framework SLSTT that composed of a Vision Transformer (Dosovitskiy et al., 2020) and a LSTM. Furthermore, MFDAN (Cai et al., 2024) introduced an additional RGB branch to construct a two-stream network (Simonyan & Zisserman, 2014), collaboratively modeling spatio-temporal features to enhance feature representation learning. Although these methods have achieved promising progress, they still rely on costly apex annotations to extract effective motion clues, which limits the effective utilization of ME data. To address this issue, we treat a ME sequence as a set of onset/near-median motion instances, which eliminates the need for apex annotations.

**Self-supervision for Micro-expression Recognition.** To alleviate data scarcity in ME research, numerous studies adopted external knowledge to improve the performance of MER models by self-supervised pre-training. SelfME (Fan et al., 2023) and SODA4MER (Zhang et al., 2025a) finetuned a first order motion model (Siarohin et al., 2019) pre-trained on the VoxCeleb dataset (Nagrani et al., 2017) to achieve self-supervised motion learning. Since the release of the large-scale ME dataset CAS(ME)[3] (Li et al., 2022), some studies have applied additional ME frames or depth information to construct self-supervised learning models. Nguyen et al. (Nguyen et al., 2023) proposed Micron-BERT, which is pre-trained on CAS(ME)[3] with specially designed modules to detect micro-movements. Li et al. (Li et al., 2025) adopted RGB and depth modalities for self-supervised contrastive pre-training, followed by fine-tuning on downstream MER tasks. Unlike these works, FRL-DGT (Zhai et al., 2023) introduced a displacement generation module that samples sufficient additional random frame pairs from the dataset to train the model in a self-supervised manner, thereby avoiding reliance on additional datasets. Similar to FRL-DGT, we do not rely on additional datasets but instead sample sufficient random instance pairs from data augmentation caused by the multiple instances representation. Through joint training of IRC and MRD modules, our method enforces the model to focus on subtle facial activation regions with diverse instance pairs as input.

## 3 PROPOSED METHOD

The framework of the proposed Ra-ICL is illustrated in Figure 2. Ra-ICL consists of three parts, i.e., Multiple Instances Representation (MIR), Instance Region Consistency (IRC) module, and Multi-Region Discovery (MRD) module. The MIR provides motion features for each ME sequence in the form of optical flow instance sets. The IRC module then takes random instance pairs as input to extract subtle motion information by enforcing attention consistency between instance pairs. Meanwhile, the MRD module uses a set of facial queries to discover meaningful facial regions, which constrains the attention on more activation regions with low intensity.

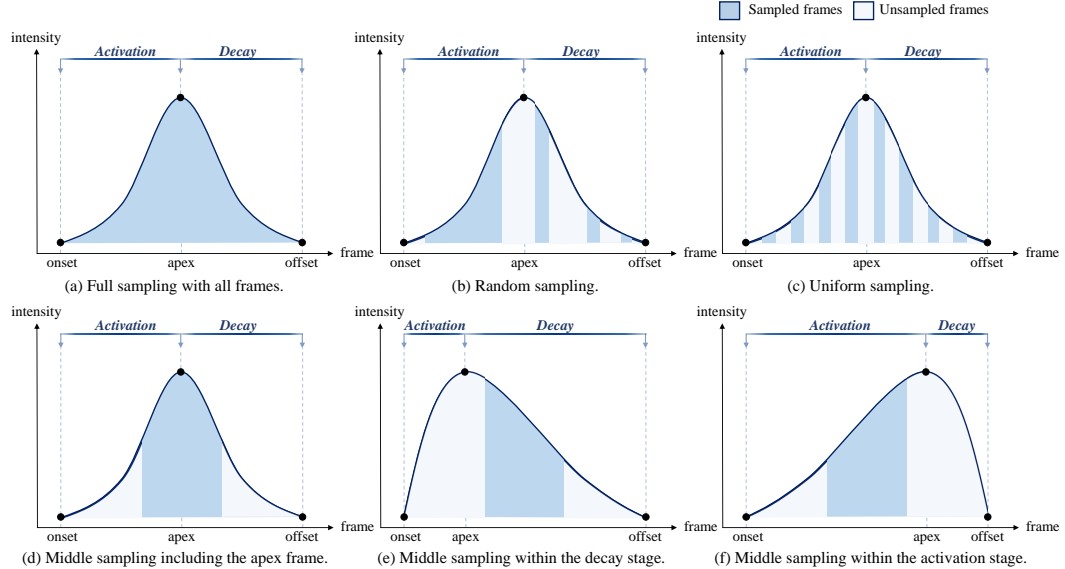

Figure 3: Different sampling strategies for Multiple Instances Representation.

### 3.1 MULTIPLE INSTANCES REPRESENTATION

The Multiple Instances Representation (MIR) forms the foundation of Ra-ICL, avoiding the need for apex annotations and enabling data augmentation. As shown in Figure 3, a complete ME movement can be divided into two stages: activation stage from onset to apex, and decay stage from apex to offset. During this process, the approximate spatial extent of facial activation regions (i.e., action units) remains consistent across the temporal domain, differing only in intensity. Therefore, frames other than the apex frame in a ME sequence also carry valid motion information with relatively lower intensity. Without using apex frame annotations, a ME sequence can be represented as a set of optical flow instances from the onset frame to arbitrary intra-sequence frames (see Figure 3(a)). These instances share similar facial activation regions, but vary in intensity.

Considering the computational cost of optical flow, the number of sampled frames $N$ must be limited rather than using all frames. Compared to random sampling (see Figure 3(b)) and uniform sampling (see Figure 3(c)), middle sampling avoids low-intensity frames typically located at both ends of the sequence. As shown in Figure 3(d), under ideal conditions, the middle sampling window covers the apex frame. When the apex frame falls outside the window, the entire sampling range lies either within the activation stage or the decay stage, as illustrated in Figure 3(e)(f). In such cases, although frames with higher motion intensity are not captured, it avoids sampling frames near the sequence boundaries where the motion magnitude is close to zero. Based on the middle sampling, the amount of motion instances is expanded from a single onset/apex pair to multiple onset/near-median pairs.

For a ME sequence, we sample $N$ consecutive frames from the middle part of the sequence. Then we obtain $N$ optical flow feature maps using the onset and the sampled $N$ frames by the TV-L1 method (Sánchez Pérez et al., 2013). The optical flow field $O_n$ between the onset frame and the $n$-th frame can be expressed as the combination of the horizontal field $u_n$ and the vertical field $v_n$:

$$O_n = \{(u_n(x, y), v_n(x, y))\}, \tag{1}$$

where $x = 1, 2, ..., H$, $y = 1, 2, ..., W$, $(x, y)$ represents the pixel position, $H$ and $W$ are the height and width of the image. Moreover, following previous works (Xu et al., 2022; Cai et al., 2024), the optical strain is further combined as the third channel to form a three-dimensional tensor similar to a RGB image. Optical strain $\epsilon_n$ is defined as the first-order derivative of the optical flow field $O_n$. The final optical flow map $I_n \in \mathbb{R}^{H \times W \times 3}$ can be expressed as:

$$I_n = [u_n, v_n, \epsilon_n]. \tag{2}$$

## 3.2 Instance Region Consistency Learning

Based on MIR, we aim to learn subtle motion dynamics from motion diversity. Given a batch of ME sequences, we randomly sample two distinct optical flow instances $I_p$ and $I_q$ to form sufficient random instance pairs for each sequence. In this way, instances with lower intensity are systematically paired with multiple instances with higher intensity in a large pool of instance pairs. We adopt the Siamese network as BYOL (Grill et al., 2020), consisting of two branches: the online network $E_\theta$ and the target network $E_\xi$. Both networks have the same residual structure (He et al., 2016), but the target parameters $\xi$ are updated with an exponential moving average (EMA) of the online parameters $\theta$ given a target decay rate $\tau \in [0, 1]$ :

$$\xi = \tau\xi + (1 - \tau)\theta. \tag{3}$$

Based on spatial consistency of motion cues, the Siamese structure learns to identify invariant activation regions from diverse instance pairs. Meanwhile, activation regions with weaker motion can be effectively identified from the guidance of instances with higher intensity. Firstly, both instances $I_p$ and $I_q$ are fed into the online network $E_\theta$ and the target network $E_\xi$. We denote $T_\theta$ and $T_\xi$ as random data transforms (e.g., color jitter and Gaussian blur) applied to $E_\theta$ and $E_\xi$, respectively. In the following sections, we consider the situation where $I_p$ is processed by $E_\theta$ and $I_q$ is processed by $E_\xi$. The spatial consistency requires the attention on activation regions for $I_p$ in $E_\theta$ aligned with that of $I_q$ in $E_\xi$, which can be realized by flip semantic consistency (Guo et al., 2019; Zhang et al., 2022b). Specifically, the attention region for image classification is flipped horizontally if the original image is flipped horizontally. Therefore, $T_\xi$ performs a certain horizontal flipping on $I_q$.

As a result, $I_\theta = T_\theta(I_p)$ and $I_\xi = T_\xi(I_q)$ are then processed by $E_\theta$ and $E_\xi$ separately to generate feature maps. The feature maps are extracted from the last convolutional layer, denoted as $\mathbf{F}_e \in \mathbb{R}^{C \times H \times W}$ with $e \in \{\theta, \xi\}$, where $C, H, W$ are the number of channels, height and width of the feature map, respectively. A subsequent global average pooling (GAP) is performed on $\mathbf{F}_\theta$ and get features $f_\theta \in \mathbb{R}^{C \times 1 \times 1}$. The pooled features are further resized to $f'_\theta \in \mathbb{R}^C$ and put through the fully connected (FC) layer with weights $\mathbf{W} \in \mathbb{R}^{L \times C}$ to compute the classification loss:

$$\mathcal{L}_{\text{cls}} = -\log\left(\frac{e^{\mathbf{W}_y \cdot f'_\theta}}{\sum_{j=1}^L e^{\mathbf{W}_j \cdot f'_\theta}}\right), \tag{4}$$

where $\mathbf{W}_y$ is the $y$-th weight from the FC layer, $y$ is the given ground truth label, and $L$ is the number of labels. Meanwhile, the attention regions of the input instance can be derived as attention heatmaps $\mathbf{M}_j(x, y) \in \mathbb{R}^{H \times W}$ by Class Activation Mapping (CAM) (Zhou et al., 2016) to indicate the relevance of spatial location $(x, y)$ for a predicted class $j$. Formally, the mapping is computed through a weighted combination of feature maps over different channels:

$$\mathbf{M}_j(x, y) = \sum_{c=1}^C \mathbf{W}(j, c)\mathbf{F}_c(x, y), \tag{5}$$

where $\mathbf{F}_c(x, y)$ represents the activation value at spatial position $(x, y)$ of the $c$-th channel, $C$ denotes the total number of feature maps, and $\mathbf{W}(j, c)$ signifies the weight coefficient corresponding to class $j$ for the FC layer. We compute attention heatmaps $\mathbf{M}_\theta$ and $\mathbf{M}_\xi$ for $E_\theta$ and $E_\xi$ according to equation 5. Finally, attention consistency loss using the mean square difference is utilized to minimize the distance between $\mathbf{M}_\theta$ and $Flip(\mathbf{M}_\xi)$:

$$\mathcal{L}_{\text{acl}} = \frac{1}{LHW}\sum_{j=1}^L \|\mathbf{M}_{\theta j} - Flip(\mathbf{M}_{\xi j})\|_2 , \tag{6}$$

where $\mathbf{M}_{\theta j}$ and $\mathbf{M}_{\xi j}$ indicate the attention heatmaps for $I_\theta$ and $I_\xi$ respectively with label $j$. Through joint optimization of $\mathcal{L}_{\text{cls}}$ and $\mathcal{L}_{\text{acl}}$, the Siamese network learns to associate facial activation regions with corresponding emotion labels.

## 3.3 Self-supervised Multi-Region Discovery

The IRC module is guided to focus on facial activation regions related to emotional labels. However, for a classification model, only the most discriminative regions are recognized (Wei et al., 2017), and some weaker but important activation regions are discarded. This phenomenon leads to classification error on the ME data with high inter-class similarity. To address this issue, we present a Multi-Region Discovery (MRD) module to discover more meaningful facial regions for a comprehensive decision rather than relying on localized activation regions.

To automatically discover different facial regions, we use a set of facial queries to look up the whole image. Specifically, following MaskFormer (Cheng et al., 2021), a Transformer decoder followed by a MLP takes $N$ facial queries (Query) that are learnable positional embeddings and the feature map $\mathbf{F}_e$ (Key and Value) as input to predict $N$ facial mask embeddings $\mathbf{Q}_e \in \mathbb{R}^{N \times D}$. Each facial mask embedding is associated with a facial region. Then we generate $N$ corresponding heatmaps $\mathbf{S}_e \in \mathbb{R}^{N \times H \times W}$ to locate facial regions on the feature map as:

$$\mathbf{S}_e[m, u, v] = \text{sim}(\mathbf{Q}_e[m, :], \mathbf{F}_e^{dense}[*, u, v]), \tag{7}$$

where $\mathbf{F}_e^{dense} \in \mathbb{R}^{D \times H \times W}$ is the dense feature map obtained by projecting $\mathbf{F}_e$ through the projector $H_e$, and $\text{sim}(\cdot)$ is the cosine similarity function. The $m$-th heatmap $\mathbf{S}_e[m, u, v]$ quantifies the relevance of the pixel $(u, v)$ in the dense feature map $\mathbf{F}_e^{dense}$ to the $m$-th facial region indicated by $\mathbf{Q}_e[m, :]$. To prevent heatmaps of specific facial regions from dominating, $\mathbf{S}_e$ is normalized along the channel dimension via Softmax, resulting in a group of probabilistic heatmaps $\mathbf{P}_e \in \mathbb{R}^{N \times H \times W}$:

$$\mathbf{P}_e[m, u, v] = \frac{\exp(\mathbf{S}_e[m, u, v])}{\sum_{n=1}^m \exp(\mathbf{S}_e[n, u, v])}. \tag{8}$$

Each channel $\mathbf{P}_e[m, *, *] \in \mathbb{R}^{H \times W}$ represents a 2D heatmap that highlights the $m$-th facial region. Based on the spatial consistency of instance pairs, we formulate the heatmap learning process as a deep clustering problem (Caron et al., 2020) to provide learning signals. Specifically, we treat the $N$ learnable facial queries as the centers of $N$ facial region clusters. Therefore, the normalized per-pixel cluster assignment $\mathbf{P}_e[*, u, v]$ between $E_\theta$ and $E_\xi$ should keep aligned, which is measured by the cross-entropy loss:

$$CE(\mathbf{P}_\xi[*, u, v], \mathbf{P}_\theta[*, u, v]|I_p) = -\sum_{m=1}^N \mathbf{P}_\xi[m, u, v] \log \mathbf{P}_\theta[m, u, v], \tag{9}$$

where the target network provides a stable target. Notice that the equation 9 is formulated for the scenario where $I_p$ is processed by $E_\theta$. We define the symmetric self-supervised alignment loss for both augmented instances as:

$$\mathcal{L}_{\text{align}} = \frac{1}{HW} \sum_{u,v} \left( CE(\mathbf{P}_\xi[*, u, v], \mathbf{P}_\theta[*, u, v]|I_p) + CE(\mathbf{P}_\xi[*, u, v], \mathbf{P}_\theta[*, u, v]|I_q) \right). \tag{10}$$

Similar to $\mathcal{L}_{\text{acl}}$, the spatial consistency of local facial regions discovered by the MRD module should be considered. Based on the learned heatmaps $\mathbf{P}_e$, the latent representations for local facial regions are obtained through:

$$\mathbf{h}_e^m = \mathbf{P}_e[m, *, *] \otimes \mathbf{F}_e = \frac{1}{\sum_{u,v} \mathbf{P}_e[m, u, v]} \sum_{u,v} \mathbf{P}_e[m, u, v] \mathbf{F}_e[*, u, v], \tag{11}$$

where $\otimes$ denotes channel-wise Weighted Average Pooling (WAP), $\mathbf{h}_e^m \in \mathbb{R}^C$ is the corresponding latent representation of the $m$-th facial region produced with $\mathbf{P}_e[m, *, *]$. The projector $H_e$ is performed on these latent representations to obtain facial embeddings:

$$\mathbf{z}_e^m = H_e(\mathbf{h}_e^m). \tag{12}$$

The cosine similarity is used to measure the consistency of produced local facial regions between $E_\theta$ and $E_\xi$:

$$\text{sim}(\mathbf{z}_\theta, \mathbf{z}_\xi|I_p) = \frac{1}{N} \sum_{m=1}^N \text{sim}(G_\theta(\mathbf{z}_\theta^m), \mathbf{z}_\xi^m), \tag{13}$$

where $G_\theta$ is the predictor on top of the projector $H_\theta$. Similar to equation 10, equation 13 should be computed symmetrically for both augmented instances as:

$$\mathcal{L}_c = \text{sim}(\mathbf{z}_\theta, \mathbf{z}_\xi|I_p) + \text{sim}(\mathbf{z}_\theta, \mathbf{z}_\xi|I_q). \tag{14}$$

### 3.4 OVERALL OBJECTIVE

We jointly optimize the equation 15:

$$\mathcal{L} = \lambda_1 \mathcal{L}_{\text{cls}} + \lambda_2 \mathcal{L}_{\text{acl}} + \lambda_3 (\mathcal{L}_{\text{align}} + \mathcal{L}_c), \tag{15}$$

where $\lambda_1, \lambda_2, \lambda_3$ are the loss weight for balancing the classification, IRC and MRD respectively. As shown in Figure 2, both online and target networks are jointly updated during training. For inference, only the online network is utilized for classification as with BYOL (see Appendix A.1.3).

Table 1: Comparison with SOTA methods in terms of UF1(%) and UAR(%) under the CDE setting. The best results are highlighted in **bold**, while the second-best results are marked with an underline.

| Methods | Composite | | CASME II | | SAMM | | SMIC-HS | |
|---------|-----------|---|----------|---|------|---|---------|---|
| | UF1 | UAR | UF1 | UAR | UF1 | UAR | UF1 | UAR |
| LBP-TOP (Zhao & Pietikainen, 2007) | 58.82 | 57.85 | 70.26 | 74.29 | 39.54 | 41.02 | 20.00 | 52.80 |
| Bi-WOOF (Liong et al., 2018) | 62.96 | 62.27 | 78.05 | 80.26 | 52.11 | 51.39 | 57.27 | 58.29 |
| STSTNet (Liong et al., 2019) | 73.53 | 76.05 | 83.82 | 86.86 | 65.88 | 68.10 | 68.01 | 70.13 |
| RCN (Xia et al., 2019) | 74.32 | 71.90 | 85.12 | 81.23 | 76.01 | 67.15 | 63.26 | 64.41 |
| FeatRef (Zhou et al., 2022) | 78.38 | 78.32 | 89.15 | 88.73 | 73.72 | 71.55 | 70.11 | 70.83 |
| SLSTT (Zhang et al., 2022a) | 81.60 | 79.00 | 90.10 | 88.50 | 71.50 | 64.30 | 74.00 | 72.00 |
| FRL-DGT (Zhai et al., 2023) | 81.20 | 81.10 | 91.90 | 90.30 | 77.20 | 75.80 | 74.30 | 74.90 |
| MFDAN (Cai et al., 2024) | 84.53 | 86.88 | 91.34 | 93.26 | 78.71 | 81.96 | 68.15 | 70.43 |
| HTNet (Wang et al., 2024) | 86.03 | 84.75 | 95.32 | 95.16 | 81.31 | 81.24 | 80.49 | 79.05 |
| MPFNet (Ma et al., 2025) | 83.20 | 84.70 | 87.90 | 89.50 | 79.10 | 82.60 | 78.10 | 78.30 |
| CSARNet (Zhao et al., 2025) | 82.39 | 83.00 | 92.54 | 92.98 | 77.32 | 78.51 | 76.05 | 76.39 |
| Ra-ICL (ours) | **88.05** | **89.11** | **96.20** | **96.20** | **86.68** | **88.85** | **81.79** | **82.74** |

# 4 EXPERIMENTS

## 4.1 EXPERIMENTAL SETTINGS

**Datasets.** We conducted experiments on four public ME datasets: CASME II (Yan et al., 2014), SAMM (Davison et al., 2016), SMIC-HS (Li et al., 2013) and CAS(ME)³ (Li et al., 2022). Appendix A.1.1 introduces the details of datasets.

**Implementation Details.** The framework is implemented by Pytorch (Paszke et al., 2019). Both online and target networks adopt randomly initialized ResNet18 (He et al., 2016) as the backbone with target decay rate $\tau = 0.99$. The optimizer is Adam with an initial learning rate of 0.001 and a weight decay of 0.0001. We train the framework with a batch size of 32 and exponential learning rate decay with the gamma of 0.9 for 100 epochs. For MIR, the number of sampled frames $N$ is set to 16, which is determined by the frame rate prior. For MRD, the number of facial queries is set to 8. In equation 15, we take $\lambda_1 = \lambda_2 = \lambda_3 = 0.5$. Appendix A.1.2 introduces the details of determining the $N$ of MIR, while Appendix A.2 provides relevant hyper-parameter analysis.

**Evaluation Protocols.** To evaluate the model performance on CASME II, SAMM and SMIC-HS, we adopt the Composite Database Evaluation (CDE) (See et al., 2019) with Leave-one-subject-out (LOSO) cross-validation to ensure a fair comparison. The CDE setting combines samples from CASME II, SAMM and SMIC-HS into a composite dataset for training. For the evaluation on CAS(ME)³, 3-class, 4-class and 7-class evaluation with LOSO are reported following previous works (Li et al., 2022; Nguyen et al., 2023). More details are displayed in Appendix A.1.4.

**Metrics.** As per standard (See et al., 2019), we adopt the Unweighted F1-score (UF1) and Unweighted Average Recall (UAR) to assess model performance. Compared to Accuracy (Acc), UF1 and UAR provide a more balanced judgement on all classes.

Table 2: Comparison with other SOTA methods in terms of UF1 (%) and UAR (%) on the CAS(ME)³ dataset. The best results are highlighted in **bold**, while the second-best results are marked with underline.

| Methods | Classes | UF1 | UAR |
|---------|---------|-----|-----|
| STSTNet (Liong et al., 2019) | 3 | 37.95 | 37.92 |
| RCN (Xia et al., 2019) | 3 | 39.28 | 38.93 |
| FeatRef (Zhou et al., 2022) | 3 | 34.93 | 34.13 |
| $\mu$-BERT (Nguyen et al., 2023) | 3 | 56.04 | 61.25 |
| HTNet (Wang et al., 2024) | 3 | 57.67 | 54.15 |
| Lite-Point-GCN (Zhang et al., 2025b) | 3 | 68.19 | 74.12 |
| Ra-ICL (ours) | 3 | **75.85** | **74.54** |
| Baseline (Li et al., 2022) | 4 | 29.15 | 29.10 |
| Baseline(+Depth) (Li et al., 2022) | 4 | 30.01 | 29.82 |
| $\mu$-BERT (Nguyen et al., 2023) | 4 | 47.18 | 49.13 |
| Lite-Point-GCN (Zhang et al., 2025b) | 4 | 47.64 | 53.66 |
| Ra-ICL (ours) | 4 | **61.03** | **58.48** |
| Baseline (Li et al., 2022) | 7 | 17.59 | 18.01 |
| Baseline(+Depth) (Li et al., 2022) | 7 | 17.73 | 18.29 |
| $\mu$-BERT (Nguyen et al., 2023) | 7 | 32.64 | 32.54 |
| Lite-Point-GCN (Zhang et al., 2025b) | 7 | 35.64 | 41.59 |
| Ra-ICL (ours) | 7 | **44.32** | **43.17** |

Table 3: Ablation study of key components of Ra-ICL. "→X" indicates replacing the corresponding component with X. The best results are **bolded**, and the second-best results are underlined.

| Method | MIR | IRC | MRD | Composite | | CASME II | | SAMM | | SMIC-HS | |
|---|---|---|---|---|---|---|---|---|---|---|---|
| | | | | UF1 | UAR | UF1 | UAR | UF1 | UAR | UF1 | UAR |
| M1 | →Apex* | × | × | 83.22 | 82.64 | 94.78 | 95.73 | 81.43 | 80.39 | 75.13 | 74.40 |
| M2 | ✓ | × | × | 84.91 | 83.72 | 93.83 | 93.36 | 83.68 | 82.76 | 78.70 | 77.86 |
| M3 | →Apex* | ✓ | ✓ | 87.61 | 86.89 | 96.18 | 96.49 | 85.60 | 82.33 | 81.45 | 81.18 |
| M4 | ✓ | × | ✓ | 86.18 | 85.92 | 96.14 | 95.54 | 81.47 | 80.90 | 79.78 | 79.77 |
| M5 | ✓ | ✓ | × | 85.81 | 85.46 | 95.77 | 95.16 | 82.53 | 79.41 | 79.28 | 79.52 |
| M6 (ours) | ✓ | ✓ | ✓ | 88.05 | **89.11** | 96.20 | 96.20 | 86.68 | **88.85** | 81.79 | **82.74** |
| M7 | →Apex | ✓ | ✓ | 87.73 | 88.73 | **97.26** | **98.48** | 85.73 | 87.89 | 80.61 | 81.02 |
| M8 | →Full | ✓ | ✓ | **88.26** | 88.21 | 96.24 | 95.16 | **88.29** | 85.47 | 81.76 | 82.45 |
| M9 | →Random | ✓ | ✓ | 87.89 | 87.87 | 95.42 | 95.45 | 86.36 | 82.91 | **82.09** | 82.62 |
| M10 | →Uniform | ✓ | ✓ | 86.64 | 86.92 | 97.21 | 97.82 | 83.34 | 81.84 | 79.13 | 79.70 |

Apex*: Using a single onset/apex motion instance. Apex: Sampling within an apex neighborhood. Full: Full sampling. Random: Random sampling. Uniform: Uniform sampling.

## 4.2 Comparison with State-of-the-art Methods

We compared the proposed Ra-ICL with existing state-of-the-art (SOTA) methods under the CDE setting. The classification results on the composite dataset, CASME II, SAMM, and SMIC-HS are reported. As shown in Table 1, both UF1 and UAR of our framework on the composite dataset are higher than 88.00%, surpassing the second-best method by 2.02% in UF1 and 2.23% in UAR. Notably, on the SAMM, our method outperforms the second-best method by 5.37% in UF1 and 6.25% in UAR. Ra-ICL also achieves the best performance on CASME II and SMIC-HS datasets. Table 5 presents classification results on the CAS(ME)$^3$ of Ra-ICL with SOTA methods. Ra-ICL achieves the best performance in all evaluation settings, demonstrating substantial improvements compared to previous approaches. It is worth mentioning that Ra-ICL does not use any apex annotations.

We attribute the superiority of Ra-ICL to its diverse motion representations and the effective exploitation of this diversity by IRC and MRD. Although previous methods adopted well-designed classification models, their performance was still restricted by the amount of onset/apex pairs. MIR represents the ME sequence by diverse motion instances to achieve effective utilization of ME data, providing enough motion cues for subsequent classification models. With the ability to capture weaker motion information, IRC and MRD learn more robust features from diverse motion cues.

## 4.3 Ablation Study

To evaluate the contribution of MIR, IRC and MRD, we conducted the ablation study under the CDE setting. Table 3 quantitatively demonstrates the contributions of each design compared to the method M6 (ours) with all designs.

**Multiple Instances Representation.** M1 utilizes a single onset/apex instance to represent the entire ME sequence, while M2 is based on MIR with middle sampling. M3 is similar to M1 but with IRC and MRD. The improvement from M1 to M2 and M3 to M6 demonstrates the superiority of MIR, indicating that the diversity of motion cues is more important than the intensity for classification.

We also conducted experiments for different sampling strategies of MIR from M7 to M10. M7 samples 16 frames from a window centered on the apex frame. M8 samples all frames within the ME sequence. M9 and M10 perform random sampling and uniform sampling with 16 frames, respectively. M6, M7 and M8 exhibit comparable performance, while M6 avoids using apex frames as M7 and reduces the computational cost of optical flow as M8. Compared to M6, M7 employs a relatively fixed sampling range, resulting in weaker motion diversity, and thus does not outperform M6. Although M8 provides greater motion diversity, it simultaneously introduces additional noise that may obscure the perception of subtle movements. M9 and M10 also achieve competitive results, indicating that MIR with randorm or uniform sampling also carries diverse motion cues.

**Instance Region Consistency Module.** From M2 and M5, we find that IRC extracts effective subtle motion cues from MIR. In addition, M4 relies solely on MRD to discover meaningful activation regions in a self-supervised manner, which lacks supervision from ground-truth labels as with M6.

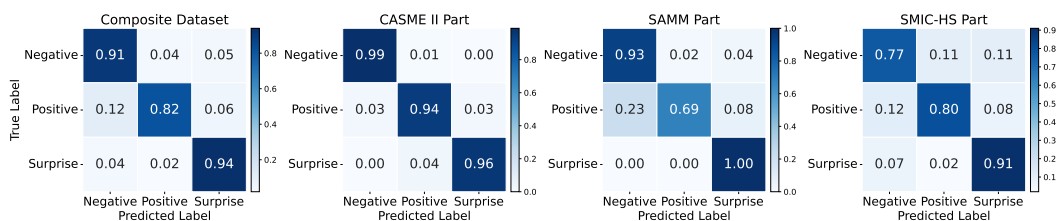

Figure 4: The confusion matrices under the CDE setting.

**Multi-Region Discovery Module.** The comparison between M2 and M4 indicates the ability of MRD to discover meaningful motion cues. Meanwhile, the improvement from M5 to M6 shows that MRD is helpful in expanding the attention of the model to encompass more subtle motion patterns.

## 4.4 VISUALIZATION

**Visualization of Confusion Matrices.** Figure 4 illustrates the confusion matrices under the CDE setting. We observe that the error primarily comes from misclassifying positive as negative. This phenomenon may stem from the bias of the model due to the largest amount of negative data.

**Visualization of Attention Heatmaps.** We conducted visualizations of attention heatmaps on a ME sequence using Grad-CAM (Selvaraju et al., 2017), as shown in Figure 5. Each row corresponds to a motion instance that shares three consistent activation regions: A, B, and C. Column (a) shows that the intensity of region C increases from row 1 to row 3, while the intensity of regions A and B keeps stable across three rows.

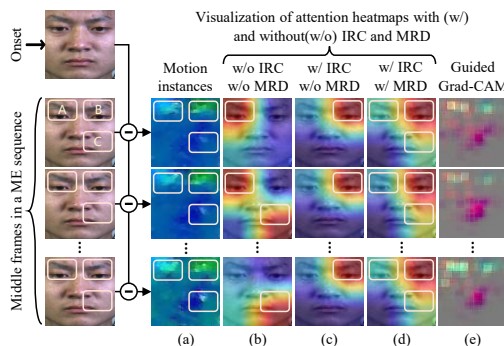

Figure 5: Visualization of attention heatmaps.

In column (b), attention heatmaps on three motion instances cover different activation regions. As the intensity of region C increases from row 1 to row 3, the attention heatmaps gradually overfit to region C. In column (c), attention heatmaps on three instances focus on the consistent region B. This comparison indicates that IRC effectively enforces visual attention consistency on different motion instances within the same instance set.

Although IRC remains attention on similar regions across different instances in column (c), regions A and C are not recognized. Consequently, column (d) combines MRD into the model to discover more meaningful regions. We observe that the model is guided to allocate more attention on regions A and C, while keeping most attention on region B.

To further visualize precise pixel-level activation regions that contribute to classification, we fuse Guided Backpropagation (GB) (Springenberg et al., 2014) and Grad-CAM via point-wise multiplication to generate Guided Grad-CAM (Selvaraju et al., 2017). As shown in column (e), pixels in regions A, B, and C contribute to the classification together. Appendix A.3 provides more visualizations of confusion matrices and attention heatmaps.

## 5 CONCLUSION

In this paper, we rethought the necessity of apex annotations and proposed an apex-free framework Ra-ICL for MER. MIR advanced the motion feature of MEs by using an instance set instead of a single motion instance. Based on MIR, IRC leveraged spatial consistency across motion instances to capture effective motion cues. Meanwhile, MRD further expanded the attention on more subtle activation regions that are easily neglected. Extensive experiments on four ME datasets demonstrated the superiority of Ra-ICL. In the future, the effectiveness of MIR deserves further investigation to promote practical applications of MER models.

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

# A APPENDIX

## A.1 MORE EXPERIMENTAL DETAILS

### A.1.1 DATASETS

We conducted experiments on four public ME datasets: CASME II (Yan et al., 2014), SAMM (Davison et al., 2016), SMIC-HS (Li et al., 2013) and CAS(ME)[3] (Li et al., 2022).

**CASME II** provides 255 samples collected from 26 subjects. These samples are classified into five categories, including Happiness, Repression, Surprise, Disgust, and Others. Videos are at 200 frames per second (fps).

**SAMM** contains 159 samples from 32 participants with eight categories, including Happiness, Surprise, Anger, Disgust, Sadness, Fear, Contempt, and Others. The frame rate is 200 fps.

**SMIC-HS** consists of 164 samples from 16 subjects classified into three categories: Positive, Negative, and Surprise. The samples are captured at 100 fps.

**CAS(ME)[3]** part A comprises 860 samples from 100 subjects. These samples are classified into seven categories: Happiness, Anger, Fear, Disgust, Surprise, Sadness, and Others. The frame rate is set at 30 fps.

### A.1.2 DETERMINATION OF THE NUMBER $N$ FOR MIR

For MIR with middle sampling, we divide the ME sequence into three segments of equal length: the initial segment, the middle segment, and the end segment. The middle segment corresponds to the sampling window. Therefore, the number of sampled frames $N$ (i.e., the length of the middle segment) can vary across different ME sequences. If the length of a ME sequence is $T$, then the number $N$ can be obtained by:

$$N = \frac{T}{3}. \tag{16}$$

To simplify the construction of instance sets, we set a unified value of $N$ for all sequences based on the frame rate prior of datasets. Specifically, the duration of a ME sequence is typically less than 0.5 seconds. Denote the frame rate as $f$, the number $N$ should satisfy the equation 17:

$$N \leq \frac{1}{3} \cdot 0.5 \cdot f = \frac{f}{6} \tag{17}$$

Under the CDE setting, the lowest frame rate is 100 fps from the SMIC-HS dataset. Therefore, setting $N$ to 16 is a reasonable choice, as it ensures motion diversity while satisfying the requirement of the equation 17. For the CAS(ME)[3] dataset, although its frame rate is 30 fps, sequence lengths vary from just a few frames to over 100 frames, which do not strictly adhere to the constraint of less than 0.5 seconds. For consistency, we also set $N$ to 16 on the CAS(ME)[3] dataset. For all datasets, we apply full sampling (i.e., all frames are used) on sequences containing fewer than 16 frames.

### A.1.3 EXPERIMENTAL CONFIGURATIONS

All experiments were performed on a high-performance computer with 16 CPU cores, 1 NVIDIA 3090 Ti GPU card, and 32 GB memory. Both online and target networks are not pretrained. Their parameters are randomly initialized. For inference, we choose the middle instance in an instance set as input to the online network. The reason for selecting middle instances is similar to that of middle sampling: to avoid using instances with excessively low motion intensity for inference.

### A.1.4 EVALUATION PROTOCOLS

For CASME II, SAMM and SMIC-HS, we combine them into a composite dataset under the CDE setting proposed by the MEGC2019 challenge (See et al., 2019) to ensure a fair comparison. This

Table 4: Details of three datasets under the CDE setting.

| Category | CASME II | SAMM | SMIC-HS | Full |
|---|---|---|---|---|
| Negative | 88 | 92 | 70 | 250 |
| Positive | 32 | 26 | 51 | 109 |
| Surprise | 25 | 15 | 43 | 83 |
| Total | 145 | 133 | 164 | 442 |

Table 5: Details of 3-class, 4-class and 7-class evaluation on CAS(ME)$^3$.

| 3-class | | 4-class | | 7-class | |
|---|---|---|---|---|---|
| Negative | 438 | Negative | 438 | Anger | 55 |
| | | | | Fear | 82 |
| | | | | Disgust | 245 |
| | | | | Sadness | 56 |
| Positive | 49 | Positive | 49 | Happiness | 49 |
| Surprise | 183 | Surprise | 183 | Surprise | 183 |
| \ | \ | Others | 131 | Others | 131 |
| All | 670 | All | 801 | All | 801 |

challenge aims to create a more realistic emotion recognition scenario by expanding the diversity of data to support data-driven deep learning. To reduce ambiguity from different stimuli and settings, emotion categories are simplified into three categories as those in SMIC-HS: **Negative** {"Repression", "Disgust", "Anger", "Sadness", "Fear", "Contempt"}, **Positive** {"Happiness"} and **Surprise** {"Surprise"}. Samples of "Others" are excluded. The final distribution with a total of 442 samples under the CDE setting is shown in Table 4. These 442 samples belong to 68 subjects. CASME II, SAMM and SMIC-HS contain 24, 28 and 16 subjects, respectively. Therefore, the Leave-one-subject-out (LOSO) cross-validation should be repeated 68 times. In each evaluation, each subject serves as the testing set, while the remaining subjects form the training set.

For the CAS(ME)$^3$ dataset, the evaluation is conducted individually for a fair comparison. 3-class, 4-class and 7-class evaluation with LOSO are reported following previous works (Li et al., 2022; Nguyen et al., 2023). The officially provided labels contain seven categories: Happiness, Anger, Fear, Disgust, Surprise, Sadness, and Others. We conducted a 7-class evaluation on these seven categories. Similarly, we can simplify the seven categories into four categories: Negative, Positive, Surprise, and Others. These four categories can be used in 4-class evaluation. Furthermore, samples labeled as "Others" can be omitted to conduct a 3-class evaluation. Details of 3-class, 4-class and 7-class evaluation are shown in Table 5. Some samples with annotation errors are excluded.

### A.1.5 METRICS

To provide a more balanced judgement on all classes, we use Unweighted F1-score (UF1) and Unweighted Average Recall (UAR) instead of the standard Accuracy (Acc). The UF1 is defined as:

$$\text{F1}_c = \frac{2 \cdot \text{TP}_c}{2 \cdot \text{TP}_c + \text{FP}_c + \text{FN}_c} \tag{18}$$

$$\text{UF1} = \frac{1}{C} \sum_c \text{F1}_c \tag{19}$$

where $\text{TP}_c$, $\text{FP}_c$ and $\text{FN}_c$ are true positives, false positives, and false negatives for class $c$, respectively. $C$ is the total number of classes. The UAR is defined as:

$$\text{UAR} = \frac{1}{C} \sum_c \frac{\text{TP}_c}{n_c} \tag{20}$$

where $n_c$ is the number of samples of the $c$-th class.

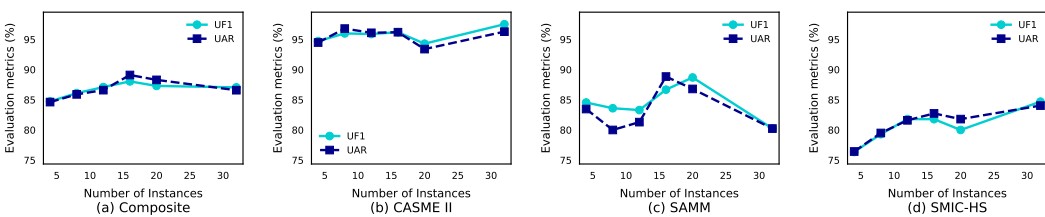

Figure 6: Hyper-parameter analysis on the number of instances under the CDE setting.

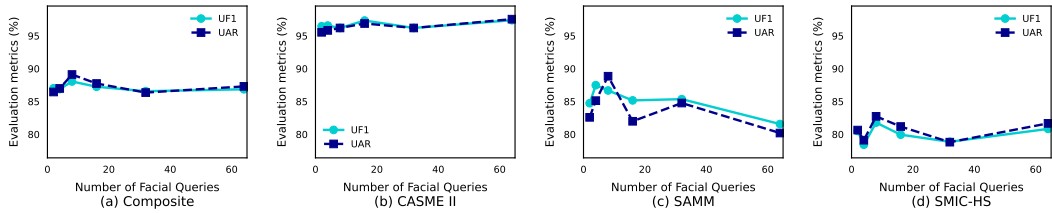

Figure 7: Hyper-parameter analysis on the number of facial queries under the CDE setting.

## A.2 HYPER-PARAMETER ANALYSIS

### A.2.1 INFLUENCE OF DIFFERENT NUMBERS OF INSTANCES

We conduct experiments on different numbers of motion instances. Considering that MEs are usually recorded below or equal to 200 fps, and ME sequences generally last less than 0.5 seconds, we selected 4, 8, 12, 16, 20, and 32 as different number of instances for a sequence. When the number of instances is fewer than 16, the middle sampling window is compressed inward. When it exceeds 16, the window expands outward toward both ends of the sequence. If the required number of instances exceeds the sequence length, full sampling is performed.

As shown in Figure 6, the model achieves the best performance when the number of instances is 16. When the number of instances is less than 16, the motion information becomes insufficiently diverse, resulting in less robust features. In contrast, when the number of instances is larger than 16, the model exhibits a slight decline in overall performance. Although increased instances provide greater motion diversity, they simultaneously introduce additional noise that may obscure the ability of the model to perceive subtle micro-movements. Therefore, it is not appropriate to increase the number of instances excessively, as it necessitates a careful trade-off between motion diversity and noise introduction.

### A.2.2 INFLUENCE OF DIFFERENT NUMBERS OF FACIAL QUERIES

MRD is enforced to discover distinct facial regions by different facial queries. Figure 7 shows the effect of different numbers of facial queries. The optimal number of facial queries is 8. If the number of facial queries is smaller, the model is at risk of failing to identify complete and meaningful facial activation regions. Conversely, if the number of facial queries is excessively large, the model may overfit to non-discriminative motion noise. This would impair its discriminative capability for critical ME features.

### A.2.3 INFLUENCE OF DIFFERENT LOSS WEIGHTS

The overall objective contains three loss weights $\lambda_1, \lambda_2, \lambda_3$ for the balance of classification, IRC, and MRD, respectively. To show the influence of these three loss weights on performance, we evaluated them from 0.1 to 2 under the CDE setting.

Figure 8 shows that the optimal $\lambda_1$ is around 0.5. When $\lambda_1$ approaches zero, the guidance from ground-truth labels reduces, resulting in a decrease in performance. As $\lambda_1$ increases, classification takes precedence, compromising the effectiveness of both IRC and MRD modules and leading to degraded performance.

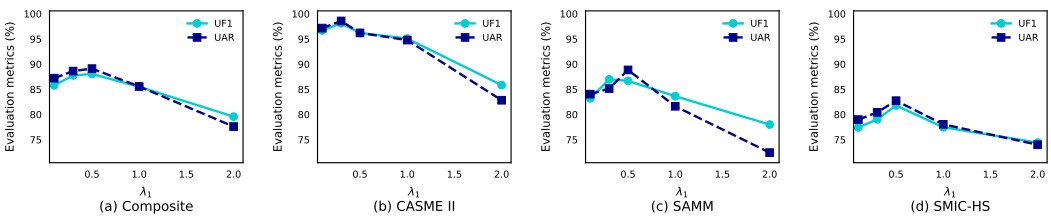

Figure 8: Hyper-parameter analysis on $\lambda_1$ under the CDE setting.

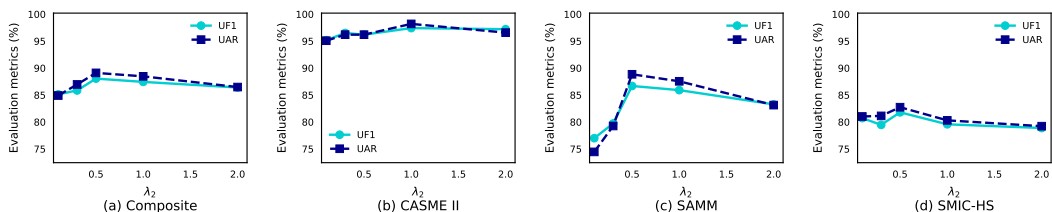

Figure 9: Hyper-parameter analysis on $\lambda_2$ under the CDE setting.

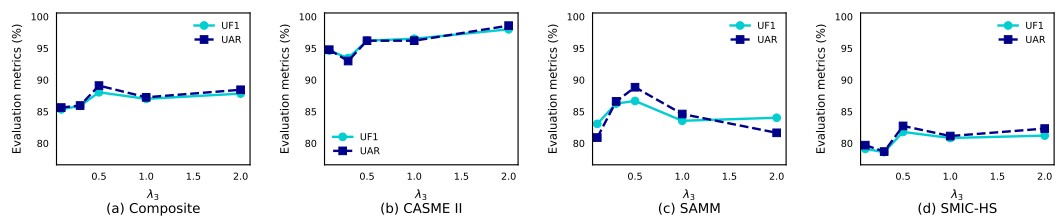

Figure 10: Hyper-parameter analysis on $\lambda_3$ under the CDE setting.

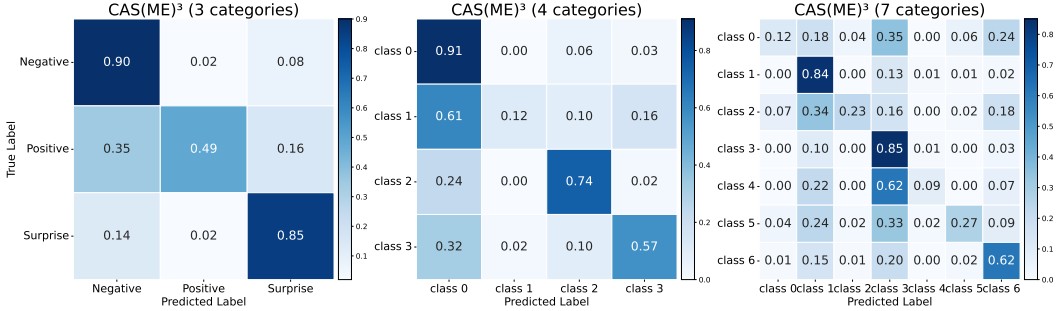

Figure 11: The confusion matrices on the CAS(ME)$^3$ dataset (From left to right: 3-class evaluation, 4-class evaluation, 7-class evaluation). For the 4-class evaluation, labels $\{0, 1, 2, 3\}$ correspond to $\{$negative, positive, surprise, others$\}$. For the 7-class evaluation, labels $\{0, 1, 2, 3, 4, 5, 6\}$ correspond to $\{$happiness, surprise, sadness, disgust, fear, anger, others$\}$.

Figure 9 demonstrates that the optimal value of $\lambda_2$ is approximately 0.5. The performance first increases along $\lambda_2$ and then decreases. If $\lambda_2$ is too small, the model will overfit to certain activation regions. If $\lambda_2$ is set larger than 0.5, the performance decreases slightly since the attention consistency loss outweighs the classification loss.

Figure 10 illustrates that the model achieves the best performance when $\lambda_3$ is equal to 0.5. When $\lambda_3 < 0.5$, the MRD module does not sufficiently constrain the model to identify more meaningful regions. When $\lambda_3 > 0.5$, it disrupts the learning process of both classification and IRC.

## A.3 VISUALIZATION

### A.3.1 VISUALIZATION OF CONFUSION MATRICES

Figure 11 shows the confusion matrices of Ra-ICL on the CAS(ME)$^3$ dataset. For the confusion matrix with 3 categories, we find that the model performs better on the negative and surprise classes but worse on the positive class. In the confusion matrix with four categories, we also observed that the positive class performed poorly. This may stem from the fewest samples of the positive class in the dataset, leading to biased model predictions. Meanwhile, the result shown in the confusion matrix with seven categories demonstrates that the model performs better in the surprise, disgust, and others classes. The three classes correspond to the three classes with the highest number of samples in the CAS(ME)$^3$ dataset.

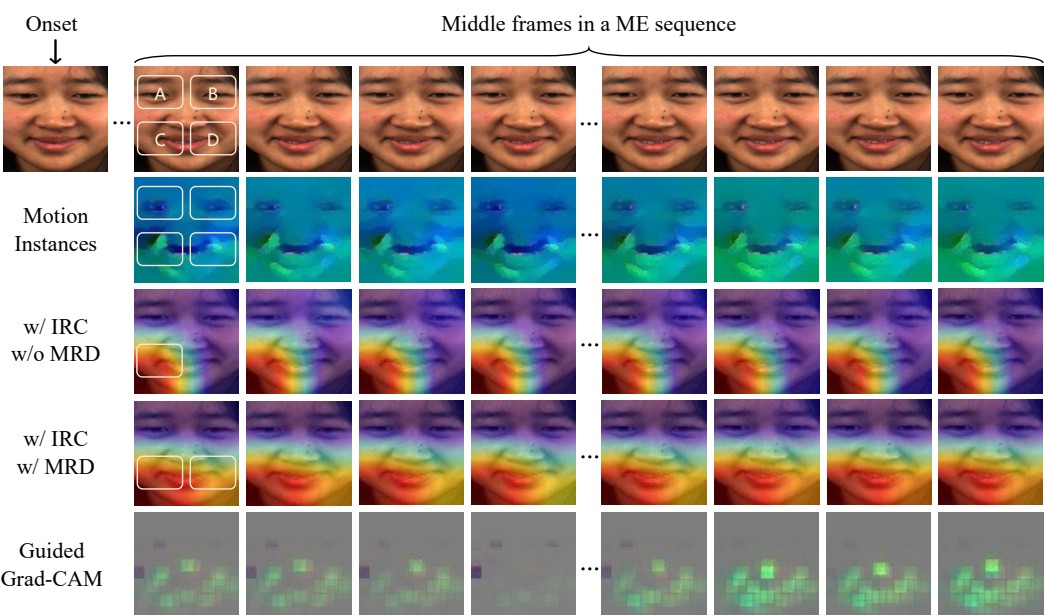

Figure 12: The visualization of attention heatmaps. The first row: RGB frames of a ME sequence. The second row: Motion instances. The third row: Attention heatmaps of the model with (w/) the IRC module but without (w/o) the MRD module. The fourth row: Attention heatmaps of the model with both IRC and MRD modules. The fifth row: Guided Grad-CAM of each motion instance.

### A.3.2 VISUALIZATION OF ATTENTION HEATMAPS

We conducted visualizations of attention heatmaps on a ME sequence using Grad-CAM (Selvaraju et al., 2017). Figure 12 shows a ME sequence with four facial activation regions: A, B, C, and D. The third row illustrates the attention regions of the model when equipped with the IRC module but without the MRD module. It is evident that the model consistently focuses on region C across all motion instances, indicating that IRC effectively enforces visual attention consistency for all instances within an instance set. However, without MRD, the attention region fails to cover region D. The fourth row represents the attention regions when both IRC and MRD modules are implemented. We observe that the model not only maintains consistent attention regions across all samples in the set but also expands its focus to the region D. Note that A and B indicate eye movements, which should be considered noise in this sample. In both the third and fourth rows, the attention of the model avoids regions A and B. This demonstrates the capability of the model to effectively distinguish between noise and subtle motions that contribute to classification. The fifth row presents the Guided Grad-CAM (Selvaraju et al., 2017; Springenberg et al., 2014) for each motion instance. It is illustrated that pixels in regions C and D are critical for classification.

## B  THE USE OF LARGE LANGUAGE MODELS

We used Large Language Models (LLMs) solely to aid and polish the writing of this paper. LLMs were not involved in the generation of research ideas, conceptualization, research design, or experimental design. The authors take full responsibility for all content presented in this manuscript, including any content generated or polished with the assistance of LLMs.

