# OpenReview forum: "Region-Aware Instance Consistency Learning for Apex-free Micro-Expression Recognition"
_ICLR.cc/2026/Conference — ICLR 2026 Conference Withdrawn Submission_

### Official Review · Reviewer_47ad · 2025-10-27

**Soundness:** 2
**Presentation:** 3
**Contribution:** 2
**Rating:** 2
**Confidence:** 4

**Summary:**

The paper proposes an apex-free paradigm for micro-expression recognition (MER), replacing the conventional onset–apex pair with Multiple Instances Representation (MIR) built from onset/near-median frame pairs. Two modules exploit these weaker but diverse motion cues: Instance Region Consistency (IRC), which enforces CAM-level attention consistency between instance pairs via a BYOL-style Siamese setup, and Multi-Region Discovery (MRD), which uses learnable facial queries and a transformer decoder to discover additional subtle regions with a self-supervised alignment/clustering loss. Experiments on several benchmarks report SOTA performance.

**Strengths:**

1. Removing apex dependence addresses both annotation cost and data under-utilization. This motivation is clear.

2. Ra-ICL tops prior methods on the composite set and each constituent dataset.

**Weaknesses:**

1. Although the paper emphasizes the proposed middle-region sampling as the core novelty, the actual gain from this strategy appears marginal. The approach essentially samples N consecutive frames around the middle portion of the video and computes optical flow between the onset frame and each selected frame. Compared with conventional strategies such as Random, Uniform, or Full sampling, the only difference lies in the temporal location of selected frames rather than a new sampling mechanism.

2. As shown in Table 3, the performance differences between the proposed method (M6) and other sampling variants (M7–M9, which do not require apex) are quite small and not statistically significant. This indicates that the sampling strategy itself contributes little to the overall improvement. The substantial performance gain of the full model seems to come mainly from the complex feature learning pipeline, including CAM-based attention consistency, Transformer-decoder region discovery, and multiple self-supervised/supervised losses, rather than from the proposed sampling policy.

3. In other words, the framework integrates several existing components (e.g., CAM, BYOL-style Siamese training, Transformer decoder) and stacks them in a heavy architecture. The reported accuracy improvements could largely be attributed to this increased model complexity and computational cost, rather than to the claimed novelty of the sampling design. Therefore, the paper’s main claimed contribution is not convincingly supported by the ablation evidence.

4. The paper claims efficiency by avoiding apex annotation, yet the overall framework seems computationally heavy.

**Questions:**

1. Why should the middle region better capture subtle micro-expressions than the entire sequence or randomly sampled regions? As shown in Table 3, both random and full sampling can also achieve a similar performance.

2. How much improvement comes solely from the sampling strategy, independent of the feature-learning modules and multi-loss design? The authors could replace the original sampling strategies in other baseline models with the proposed middle sampling and observe the corresponding performance changes.

3. What is the additional computational cost (training/inference time, FLOPs, or preprocessing) introduced by the complex modules and multiple loss terms compared with standard apex-based methods?

---

### Official Review · Reviewer_cKyk · 2025-10-30

**Soundness:** 3
**Presentation:** 3
**Contribution:** 2
**Rating:** 6
**Confidence:** 2

**Summary:**

This paper addresses the challenges of Micro-Expression Recognition (MER), which traditionally relies on labor-intensive apex frame annotations and struggles with limited data utilization. The authors propose a novel apex-free framework called Region-aware Instance Consistency Learning (Ra-ICL) to eliminate the need for apex annotations while effectively capturing subtle motion dynamics. Ra-ICL consists of three core components: (1) Multiple Instances Representation (MIR), which models a ME sequence as a set of onset/near-median motion instances instead of a single onset/apex pair; (2) Instance Region Consistency (IRC) module, which enforces visual attention consistency across different instances in the same set via a Siamese network and attention heatmaps; (3) Multi-Region Discovery (MRD) module, a self-supervised component that uses learnable facial queries to identify subtle, often neglected facial activation regions. Extensive experiments on four public ME datasets (CASME II, SAMM, SMIC-HS, and CAS(ME)³) under the Composite Database Evaluation (CDE) setting and Leave-one-subject-out (LOSO) cross-validation show that Ra-ICL outperforms state-of-the-art (SOTA) methods in terms of Unweighted F1-score (UF1) and Unweighted Average Recall (UAR), even without using any apex annotations. Ablation studies further validate the contribution of each component to the framework’s performance.

**Strengths:**

Originality：Introduces a novel apex-free MER paradigm, replacing reliance on apex annotations with a multi-instance representation (MIR) of onset/near-median frames. The combination of IRC (for attention consistency) and MRD (for self-supervised region discovery) presents a fresh integration of supervised and self-supervised learning to capture weak motion cues.
Quality：Rigorous design: MIR sampling is well-justified, IRC uses Siamese+EMA for robust attention consistency, and MRD integrates Transformer decoders with clustering loss. Comprehensive experiments on four datasets with solid ablations validate the framework's effectiveness.
Clarity：Well-structured and readable. Key components (MIR, IRC, MRD) are clearly explained, supported by intuitive figures and formulas. Experimental settings and results are thoroughly documented for easy replication.
Significance：Practically reduces annotation cost, enhancing MER's applicability in real-world scenarios. Theoretically challenges the necessity of apex frames and demonstrates effective learning from limited data without external pre-training.

**Weaknesses:**

Limited Motion Diversity Analysis: The ablation study compares sampling methods but fails to investigate how the variation in motion duration between frames affects performance. It's unclear which types of motion instances are most informative.
Insufficient Apex-Free Comparisons: As an apex-free method, the paper lacks explicit comparisons to other recent works that also avoid apex annotations. A direct comparison to an adapted FRL-DGT would better validate Ra-ICL's claimed superiority.
No Computational Cost Analysis: The computational overhead of Ra-ICL's two-branch Siamese network and Transformer is not quantified. Metrics like training time and memory usage are needed to assess practical efficiency.

**Questions:**

1. The paper sets N=16 for MIR based on frame rate priors, but how does the temporal distribution of near-median frames (e.g., frames closer to the onset vs. closer to the offset) affect the model’s ability to capture subtle motion? For example, do instances with larger temporal gaps from the onset (i.e., closer to the offset) provide more complementary information than those with smaller gaps? Could a dynamic sampling strategy (e.g., weighting instances by motion intensity) further improve performance?

2. Are there any existing MER methods that explicitly avoid apex annotations (even if not framed as “apex-free”)? If yes, how does Ra-ICL’s architecture (e.g., MIR, IRC, MRD) differ from these methods, and why does it outperform them? If no, can the authors clarify why prior work has not explored apex-free MER, and how Ra-ICL overcomes technical barriers that may have prevented this?

3. Can the authors provide quantitative data on Ra-ICL’s computational cost (e.g., training time per epoch on a NVIDIA 3090 Ti, inference time per sample, GPU memory usage) and compare it to SOTA methods (e.g., MFDAN, HTNet)? For applications requiring real-time processing, would Ra-ICL’s computational overhead be prohibitive, and are there potential optimizations (e.g., smaller backbones, simplified MRD) to reduce cost without sacrificing performance?

4. For ME sequences with extremely low motion intensity (e.g., subtle eye movements), does MIR’s middle sampling still capture sufficient motion cues? Are there cases where even multiple onset/near-median instances fail to provide discriminative information, and if so, how could Ra-ICL be adapted to handle such cases (e.g., integrating RGB texture features with optical flow)?

5. The MRD module uses 8 learnable facial queries, justified by hyper-parameter analysis. However, do these queries learn to map to specific facial regions (e.g., eyes, mouth, eyebrows) consistently across datasets, or are they dataset-specific? Can the authors visualize the regions targeted by individual queries to confirm that they correspond to meaningful facial activation areas (rather than noise)?

---

### Official Review · Reviewer_zbPm · 2025-10-31

**Soundness:** 3
**Presentation:** 2
**Contribution:** 2
**Rating:** 2
**Confidence:** 4

**Summary:**

This paper proposes a novel apex-free micro-expression recognition (MER) framework called Region-aware Instance Consistency Learning (Ra-ICL). This method aims to address the bottleneck caused by existing MER methods' heavy reliance on costly apex frame annotations. The authors' key insight is that a micro-expression sequence can be represented as a set of motion instances composed of 'onset/near-mid' frame pairs, which are consistent in spatial activation regions but differ in intensity. Based on this, the paper introduces two key modules: the Instance Region Consistency (IRC) module, which captures subtle motion dynamics by enforcing visual attention consistency across different instance pairs of the same sample; and the Multi-Region Discovery (MRD) module, which uses self-supervised learning to discover additional overlooked subtle activation regions. Extensive experiments on four public datasets demonstrate that this method outperforms existing state-of-the-art approaches without any apex frame annotations.

**Strengths:**

1. The paper accurately points out the high cost of vertex labeling in current 'start-end' pair-based MER methods and proposes a new 'vertex-free' paradigm. This motivation is very reasonable and practical.
2. Experiments were conducted on four mainstream micro-expression datasets (CASME II, SAMM, SMIC-E, CAS(ME)²), and the results fully demonstrate the effectiveness and generalization ability of the method.
3. Ablation experiments thoroughly validated the effectiveness of each module and analyzed the impact of different sampling strategies. Through visualization methods such as attention heatmaps and confusion matrices, the model's focus areas and classification performance are intuitively displayed.

**Weaknesses:**

This paper contains a promising idea and strong experimental results, but the experimental comparisons are insufficient and the methodology is not clearly described.

**Questions:**

1. Although experiments show that intermediate sampling yields the best results, in some ME sequences, the key frame may be at either end of the sequence. Is there a mechanism to handle this situation?
2. How does the model perform in terms of generalization across datasets (such as from CASME II to SAMM)? Are there any cross-dataset experiments?
3. The relationship between IRC and MRD is unclear: Are these two modules executed sequentially, in parallel, or nested? Figure 2 is too simplistic and fails to clearly show the data flow and interactions between modules. Readers have to repeatedly switch between the main text and the formulas just to barely understand it.
4. The formulas (7) to (14) in the MRD section are overloaded with symbols, but the core idea (discovering different regions through self-supervised clustering) is drowned out by the mathematical notation. It feels more like the content of the appendix has been moved into the main text, making it very hard to read.
5. The paper mentions the use of 'onset/near-median frame pairs,' but does not explain in detail how 'near-median' is defined and selected. Is it the midpoint frame of the sequence, or a dynamically chosen frame? How does this selection affect the quality of the instance and the final performance?
6. The paper identifies 'dependency on vertex frames' as a key point of criticism and uses this as the main motivation for its own method. However, this methodological limitation is widely recognized within the community, and there are already several works aimed at reducing or eliminating reliance on vertex frames (such as FRL-DGT, which is cited in the paper). The authors fail to clearly explain the fundamental differences between their method and these existing 'vertex-free' or 'weakly supervised' approaches.
7. The sentences are lengthy, and the terminology and symbols are inconsistent throughout the text, which undermines professionalism and readability. The quality of the charts needs improvement.

---

### Official Review · Reviewer_UVoD · 2025-11-01

**Soundness:** 3
**Presentation:** 2
**Contribution:** 3
**Rating:** 6
**Confidence:** 3

**Summary:**

This paper proposes an apex-free paradigm for micro-expression recognition (MER) that reduces the cost of apex annotations. The key component is Region-aware Instance Consistency Learning (Ra-ICL), which treats each sequence as a set of multiple onset vs near-median frame pairs (instances) to broadly exploit weak yet valid motion cues.

**Strengths:**

- An apex-free method for MER is proposed. In general, precise apex frame annotation imposes a heavy burden on annotators; the proposed approach can reduce annotation costs.
- Instance Region Consistency (IRC) module is proposed. IRC learns consistency of attention over facial activation regions across instances with different intensities. A BYOL-style Siamese network (online/target with EMA updates) encourages CAM consistency, including horizontal flip consistency.
- Multi-Region Discovery (MRD) module is proposed. MRD, via self-supervision, expands attention to subtler multiple activation regions that are often overlooked. In combination with IRC, it strengthens the extraction of information from weak motion.

**Weaknesses:**

- From Section 3.3, the mathematical expression of tensor operations is insufficient. Notation such as $Q_e[m, :]$ and $F^{dense}_e[, u, v]$ appears to follow Python-style array indexing, but it is not standard mathematical notation and is not clearly defined in the paper. In particular, the distinction between $:$ and $*$ is unclear. Moreover, the relationship to earlier notation such as $M_j(x, y)$ used before Section 3.2 is not well specified.
- It would be helpful to provide a more detailed qualitative analysis of how the proposed mechanism operates. Although Figure 5 visualizes attention heatmaps, analyses comparing them with the true apex, e.g., demonstrating that the learned representations sufficiently cover apex-level information, would strengthen the paper.
- There should be a direct comparison with a straightforward two-stage pipeline that chains existing micro-expression spotting and recognition.
- In addition to micro-expressions, accuracy for macro-expression recognition should also be reported. These can be handled under a similar framework, and many papers report both.

**Questions:**

As mentioned above, please provide a mathematically correct explanation of the notation for tensor operations.

---

### Note · Authors · 2025-11-14

I have read and agree with the venue's withdrawal policy on behalf of myself and my co-authors.